# Generalized Predictive Coding: Bayesian Inference in Static and Dynamic models

**André Ofner**[1]    **Beren Millidge**[2]    **Sebastian Stober**[1]

[1] AILab, Otto-von-Guericke University, Magdeburg, Germany
[2] MRC Brain Networks Dynamics Unit, University of Oxford
`{ofner, stober}@ovgu.de`   `beren@millidge.name`

## Abstract

Predictive coding networks (PCNs) have an inherent degree of biological plausibility and can perform approximate backpropagation of error in supervised learning settings. However, it is less clear how predictive coding compares to state-of-the-art architectures, such as VAEs, in unsupervised and probabilistic settings. We propose a PCN that, inspired by generalized predictive coding in neuroscience, parameterizes hierarchical distributions of latent states under the Laplace approximation and maximises model evidence via iterative inference using locally computed error signals. Unlike its inspiration it uses multi-layer neural networks with nonlinearities between latent distributions. We compare our model to VAE and VLAE baselines on three different image datasets and find that generalized predictive coding shows performance comparable to variational autoencoders trained with exact error backpropagation. Finally, we investigate the possibility of learning temporal dynamics via static prediction by encoding sequential observations in generalized coordinates of motion.

## 1   Introduction

Predictive coding is an influential theory in neuroscience that describes brain function as learning and maintaining a generative model of the world by minimising prediction errors about sensory and internal states [3, 18]. Static PCNs are organised hierarchically, where top-down signals from higher layers predict the activity of the layer below and bottom-up signals convey prediction errors. In dynamical predictive coding models each layer additionally predicts temporal changes of expected neural activity in the layer below. Given these dynamics, Hebbian weight updates can be defined that minimize the prediction error at each layer of the network.

The weight and activity update dynamics of PCNs can be interpreted as performing variational inference (VI) by iteratively refining an inferred distribution over possible causes $p(z|o)$ of observed sensory data $o$ [6, 5, 21]. In variational inference, an approximate distribution $q(z; \lambda)$ is fit to the generally intractable posterior $p_\theta(z \mid o)$ by optimizing the variational free energy $\mathcal{F}$, also known as evidence lower-bound (ELBO) [6, 7]: $\mathcal{F}_\theta(o; \lambda) = \mathbb{E}_{q(z;\lambda)} [\ln p_\theta(o, z) - \ln q(z; \lambda)]$

In predictive coding, we define $q(z; \lambda)$ to be a simple diagonal or full-covariance Gaussian distribution with $\lambda$ as the sufficient parameters, i.e. the mean and covariance. Given the generative model $\theta$ (decoder) of a particular hierarchical layer, inference in predictive coding models proceeds by estimating the optimal variational parameters $\lambda^*$ that maximize model evidence given observed data and current parameterization. Learning of the parameters of the generative model $\theta$ can be achieved by performing a gradient descent on $\mathcal{F}_\theta(o; \lambda^*)$ with respect to $\theta$ which results in Hebbian weight updates. Crucially, learning and inference in PCNs is driven by locally generated predictions and prediction errors. In hierarchical PCNs, the predicted distributions of higher layers foster empirical

4th Workshop on Shared Visual Representations in Human and Machine Visual Intelligence (SVRHM) at the Neural Information Processing Systems (NeurIPS) conference 2022. New Orleans.

priors for the next lower layer: $p(z, o) = p(o \mid z_1) p(z_1 \mid z_2) \ldots p(z_{L-1} \mid z_L)$, such that a layer's inference model can be interpreted as the next higher layer's generative model.

The variational autoencoder (VAE) is a highly influential class of deep neural networks that performs amortized inference of $\lambda$ using an inference model $\phi$ (encoder) [10]. The inference model in VAEs learns to predict the approximate posterior $q_\phi(z \mid o)$ by learning the parameters $\phi$ of the variational mapping over a dataset. In contrast to the Hebbian updates in PCNs, VAEs are trained using exact backpropagation of error through the entire model [20]. More recently, the notion of iterative inference has been adapted for VAEs to improve the posterior distribution [16, 13]. While VAEs and static PCNs have striking similarities in terms of architecture and optimisation scheme, there is still a lack of quantitative comparisons in the literature. Similarly, various deep recurrent models for predicting sequential stimuli have been developed, but lack exhaustive comparison to dynamical PCNs [2, 19, 8]. To start addressing these gaps the main contributions of this paper are:

- We compare a static PCN model with full covariance estimation via the Laplace approximation for nonlinear neural networks to VAE baselines on several image datasets

- We extend our PCN model to temporal predictions of video in generalized coordinates of motion in line with generalized predictive coding in Neuroscience [5]

## 2 Generalized predictive coding

Generalized predictive coding (GPC) describes a particularly influential class of PCNs that covers static and dynamic models in combination with generalized coordinates of state motion and the Laplace approximation [5, 6, 4, 1]. Static GPC networks infer the conditional mean and covariance of cause states $v$ and hidden states $x$. Each hierarchical layers predicts the expected activity in the next lower layer using non-linear function $g$: $y = g(x, v, \theta) + z$. Dynamical GPC networks additionally predict the motion of hidden states $\dot{x} = f(x, v, \theta) + w$ using a non-linear transition function $f^1$. $z$ and $w$ denote observation noise and transition noise respectively. While cause states are predicted hierarchically, hidden states are usually not observed by higher layers.

Under the assumption of local linearity, GPC uses states in generalized coordinates of motion $\tilde{y} = [y, y', y'', \ldots]^{\mathrm{T}}$, where $y'$ denotes the temporal derivative at $y$. Similarly, for cause and hidden states:

$$y = g(x, v) + z \qquad x' = f(x, v) + w$$
$$y' = g_x x' + g_v v' + z' \quad x'' = f_x x' + f_v v' + w'$$

Using Gaussian priors $p(z) = \mathcal{N}(z; \bar{\mu}, \Sigma)$, GPC infers posterior distributions of the causes $p(\tilde{x} \mid \tilde{v}) = N\left(D\tilde{x} : \tilde{f}, \tilde{\Sigma}^z\right)$ and the hidden states $p(\tilde{y} \mid \tilde{x}, \tilde{v}) = N\left(\tilde{y} : \tilde{g}, \tilde{\Sigma}^z\right)$. Here, $D$ denotes a derivative operator that replaces each order of state motion with the next higher order: $x \leftarrow x', x' \leftarrow x'', \ldots$.

While conditional mean parameters $\mu$ are encoded explicitly, the covariance $\Sigma$ is encoded implicitly as a function of the mean using the Laplace approximation (LA). Under the LA, the covariance is determined by the local curvature of $-\log p_\theta(y, v, x)$ at the inferred mode of $p_\theta(v, x \mid o)$. Figure 1 shows dynamical and static GPC in comparison to a VAE.

GPC proceeds by expressing the free energy $\mathcal{F}$ of each hierarchical layer $l$ as a function of precision weighted prediction errors $\xi^{(l,o)} = \Sigma^{(l,o)^{-1}} \epsilon^{(l,o)}$ for outgoing predictions and for top-down predictions $\xi^{(l,v)} = \Sigma^{(l,v)^{-1}} \epsilon^{(l,v)}$ from the next higher layer. Here, precision is the inverse of the covariance $\Sigma$. Depending on the chosen prior, the distance between prior and posterior distribution is measured by $\xi^{(l,p)} = \Sigma^{(l,p)^{-1}} \epsilon^{(l,p)}$. Here, $\epsilon^p = \mu - \bar{\mu} = \mu - 0$, is the prediction error between posterior and prior and $\epsilon^{(l)v} = \mu^{(l)} - g(\mu^{(l+1)})$ is the hierarchical prediction error between layers (or the sensory prediction error at the lowest layer). For dynamical models, the generalized predictions $\tilde{y}$ result in generalized errors $\tilde{\epsilon} = \tilde{y} - \tilde{o} = [\epsilon, \epsilon', \epsilon'', \ldots]^{\mathrm{T}}$. Inference in each layer is done via gradient descent on $\xi = \Sigma^{-1} \epsilon$ for cause states $\dot{\tilde{\mu}}^{(l)v} = \tilde{\mu}^{(l)v} - \tilde{\varepsilon}_v^{(l)T} \xi^{(l)} - \xi^{(l+1)v}$ [5]. Within each hierarchical layer, the motion of hidden states is inferred as $\dot{\tilde{\mu}}^{(l)x} = D\tilde{\mu}^{(l)x} - \tilde{\varepsilon}_x^{(l)T} \xi^{(l)}$.

---

[1]Since static models lack dynamical predictions, hidden states can usually be ignored in the static case.

An appeal of the GPC model is that it provides a simple model inversion scheme that with a strong degree of neurobiological plausibility[5, 1]. Encoding state motion in generalized coordinates casts dynamical predictions (which normally require complex recurrent connectivity) as a static prediction mechanism. Similarly, the Laplace approximation provides a simplified scheme to represent uncertainty that allows approximate inference to be performed with simple Hebbian updates.

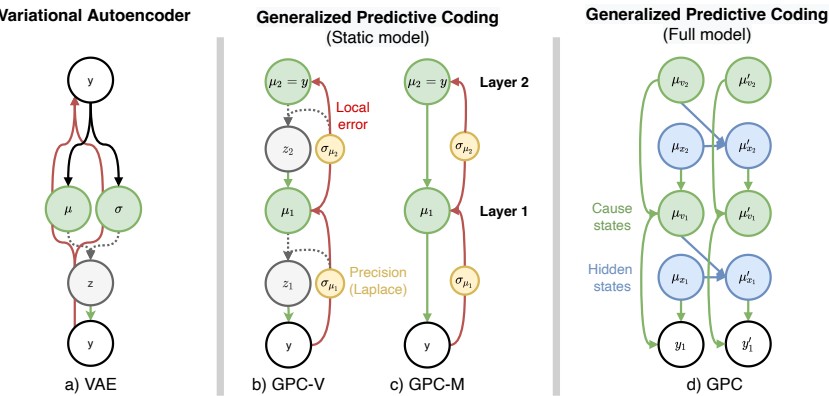

Figure 1: Variational autoencoders (a) encode mean and variance of their latent distribution. Error signals are propagated through the entire network via the backpropagation algorithm. Generalized predictive coding (b-d) propagates local errors and encodes only the mean under the Laplace approximation. The variance is a function of the mean and can be explicitly sampled (b) or appears only as error weighting terms (c).

## 3 Implemented models and baselines

### 3.1 VAE and VLAE baseline

We use the conventional VAE architecture with fully factorized normal distribution, reparameterization of the latent distribution and training via backpropagation of error [10]. It uses a single sample from the latent distribution to estimate the reconstruction error. The complexity is measured using the analytical KL divergence between inferred distribution and a standard normal prior. The VLAE is a variant of the VAE with iterative mode seeking that defines a full-covariance Gaussian posterior at the mode using the Laplace approximation [16]. The VLAE uses a single sample from the latent distribution at the inferred mode as input to the decoder. The model uses a decaying learning rate for mode seeking. For the VAE and VLAE models, the encoder and decoder consist of two ReLU activated layers with 256 hidden units and parameterize 16 latent units.

### 3.2 Static GPC model

We implement a static GPC with two hierarchical layers and fix the mean of the second layer's latents to the data. In this setup, the output of the second hierarchical layer provides empirical priors via amortized inference on the first hierarchical layer's cause states $p(v_1)$ [2]. The resulting architecture resembles that of an autoencoder as it uses the second layer's generative model as the first layer's inference model. Predictions between cause states are parameterized by a dense neural network with three layers. All generative networks have 256 hidden units and 16 latent units (causes). In contrast to the VAE baseline, inference and learning in the presented network (GPC-M) does not involve a sampling step. For comparison we also implement a network (GPC-V) that is trained using stochastic updates with a single sample from the posterior distribution. Section A of the Appendix provides a detailed explanation of the objective used for learning and inference in the proposed model.

---

[2]Note that the cause state of the second hierarchical layer is fixed, i.e. iterative inference is restricted to the first hierarchical layer.

### 3.3 Laplace approximation with ReLU nonlinearity

Inspired by the work of Park et al. [16], we employ ReLU non-linearity for the input and hidden layer, followed by a linear output layer. For network weights $W$ and Jacobian $W_z$ with respect to latent states $z = (v, x)$ ReLU non-linearity allows to efficiently compute the precision of inferred posterior states $\Sigma^{-1} = -\nabla_z^2 \log p_\theta(o, z)\big|_{z=\mu} = W_z^T W_z + I \big|_{z=\mu}$ by computing binary activation masks $\mathrm{ReLU}(Wz) = O(Wz)$ during the decoder's forward pass and recursively multiplying with the decoder's weights [16]. After a fixed amount of inference steps towards the posterior mode the approximate posterior distribution is $q(z \mid o) = \mathcal{N}(\mu, \Sigma)$ using the LA [16]. This distribution is then used to compute gradients for the weights. We perform exact error propagation to the weights strictly locally within each hierarchical layer using automatic differentiation in PyTorch [7, 17]. This is in contrast to a backpropagation pass over all parameters, such as in VAEs, where the decoder's error directly drive updates of encoder parameters.

### 3.4 Dynamical GPC model

We also implement two variants of dynamical GPC models to test amortized inference and dynamical inference of causes respectively: A simplified model with two hierarchical layers without cause states that predicts hidden states top-down. Again, the data serves as fixed input to the second hierarchical layer. The second model consists of a single hierarchical layer that infers cause and hidden states with associated dynamics using multiple shooting, as explained in the next section. We use 16 units for cause and hidden states in all models. Differently to the static model, we do not employ biases and use weighted errors at each inference step for the second model, as detailed in Appendix A.

Dynamical GPC models are trained like static GPC via iterative inference. However, in addition to decoding the states $y = g(z)$, the explicitly represented state motion $\tilde{y} = g(\tilde{z})$ is decoded and compared with the data $\tilde{o}$. During decoding the states $y = g(z)$, the Jacobian $W_z$ at the currently inferred mode is computed by masking the decoder network. All other orders of state motion $y' = g_z z', y'' = g_z z'', ...$ are decoded through this masked decoder network $W_z$. Similarly, during the prediction of hidden state motion $x' = f(z)$ the Jacobian of the transition network $f_z$ is computed. This Jacobian is reused for all higher order hidden state motion predictions $z'' = f_z(z')$, $..., z^N = f_z(z^{N-1})$. We use $N = 2$ for the first and $N = 3$ for the second GPC model. Amortized inference on the hidden states and their motion is possible by repeating the process with the second hierarchical layer's decoder. We interpret the Taylor series expansion underlying the forward and inverse embedding of sequential data as a convolution operation along the temporal axis, which can efficiently be computed using convolutional kernels. More details and examples for such kernels are shown in section B of the Appendix. Figure 3 a) displays the dynamical connectivity in a hierarchical GPC layer.

### 3.5 Multiple shooting

Following related work on multiple shooting (MS) based training of Neural ODEs we train the model by splitting discrete sequences into multiple segments, which are optimised in parallel [23, 22]. We sample discrete sequences $[o_{t_1}, ... o_{t_n}]$ of length $n$ at $m$ shooting points $[o_{\tau_1}, ... o_{\tau_m}]$ which are then embedded into generalized coordinates. For $b$ sequences sampled at $m$ point, the network is trained with a batch size of $b * m$. In practise, this means that we can omit the term $D\tilde{\mu}^{(l)x}$ for the dynamical predictions, which would be required for sequential filtering. Figure 2 shows an example with two discrete sequences. Crucially, while hidden states are inferred for each shooting point $o_{\tau_i}$ individually, the prediction errors for cause states are averaged over all $m$ samples from a sequence. We use MS with $b = 4, m = 8$ for the dynamical GPC model that infers causes and $b = 64, m = 1$ otherwise.

## 4 Static predictive coding on MNIST, FashionMNIST and OMNIGLOT

We train and evaluate static models on MNIST [12], FashionMNIST [24] and OMNIGLOT [11]. Unlike the VLAE baseline, we do not initialise the decoder output variance based on dataset statistics. Instead we add noise from a uniform distribution and apply a logit transformation for all datasets. Tables 1 and 2 shows test results on all datasets for 3 iterative inference steps using the conventional train and test splits. Listed are mean and standard deviation across 5 runs. We trained for $1e + 4$ steps with the ADAM optimiser at a learning rate of 1e-3 and a inference update rate of 0.01 for GPC-M

and GPC-S [9]. The VLAE is trained with the default inference update rate of 0.5 [16]. GPC-S and GPC-M slightly outperform the VAE on MNIST and OMNIGLOT, while the VLAE model consistently outperforms both PCNs and VAEs on all datasets. This indicates that PCNs, despite lacking exact error signals for the inference network learn a generative model that is comparable in performance to VAEs. The GPC-M model, without explicit sampling, outperformed the sample-based GPC-S model, except for one configuration on MNIST. In terms of divergence from the prior, the GPC models consistently showed posterior complexity that is comparable to, but slightly higher than, VAE complexity. For more details on model complexity and the influence of inference steps, see Section C of the Appendix. For all tested models, increasing the number of inference steps is beneficial only for low numbers of steps. We found that reducing the inference learning rate or adding a decay term can improve stability, but did not include it in our experiments.

|       | MNIST      | fMNIST     | OMNIGLOT   |
|-------|------------|------------|------------|
| GPC-S | 748.7±3.7  | 722.9±2.5  | 994.8±2.9  |
| GPC-M | 756.4±5.0  | 714.9±1.1  | 990.8±1.8  |
| VAE   | 759.5±3.9  | 713.1±1.6  | 995.1±1.6  |
| VLAE  | **713.5±3.7** | **683.6±1.3** | **956.6±2.4** |

Table 1: Negative evidence lower bound (test set)

|       | MNIST      | fMNIST     | OMNIGLOT   |
|-------|------------|------------|------------|
| GPC-S | -699.6±3.8 | -678.0±2.6 | -952.0±2.4 |
| GPC-M | -707.5±5.2 | -665.9±1.3 | -953.7±1.8 |
| VAE   | -712.9±4.2 | -671.5±1.5 | -958.5±1.7 |
| VLAE  | **-666.8±3.9** | **-641.5±1.4** | **-918.5±2.6** |

Table 2: Accuracy (test set)

## 5 Dynamical predictive coding on the dSprites dataset

To assess the capabilities of dynamical PCNs we train a dynamical GPC model on a variant of the Disentanglement testing sprites dataset [15]. Most conventional video benchmarks have relatively low sampling rates, where the local linearity assumption does not hold. We generate high resolution videos for a single direction of rotation (counterclockwise) and use random, but constant, values for the remaining latent factors. We applied Gaussian blur to all images and cropped the videos to 32x32 pixel resolution, making sure that no sprites appear outside the area.

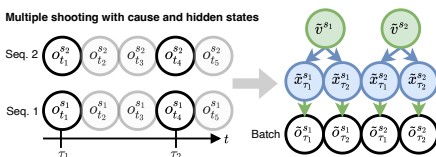

Figure 2: Multiple shooting

|         | GPC-all       | GPC-L1      |
|---------|---------------|-------------|
| MSE     | **0.432±0.124** | 0.476±0.204 |
| Layer 1 | **0.768±0.257** | 0.779±0.422 |
| Layer 2 | **0.097±0.013** | 0.173±0.031 |

Table 3: Accuracy (dSprites)

Table 3 shows mean and standard deviation of the mean squared error (MSE) for 10 runs over 3e+4 updates using two different variants of the simplified dynamical model: The GPC-all model was trained using the prediction error from both dynamical layers, while GPC-L1 only considers the error in the lowest dynamical layer. Both models smoothly predict the constant rotation across latent factors. GPC-all shows improved MSE in terms of total and per-layer prediction. This indicates that including higher-order dynamical predictions errors propagated through the network's Jacobian indeed improves accuracy. We found that GPC-L1 reacts poorly to increased latent dimensionality and stops predicting any meaningful state motion when 32 or more latent units were used. In contrast, GPC-all showed meaningful transitions for larger embeddings.

## 6 Simultaneously inferring cause and hidden states

We found that training a dynamical model that infers causes and hidden states simultaneously on the rotating dSprites dataset leads to a clear clustering of causes into the two directions of motion for the inferred cause states, as shown in Figure 4 b). The hidden states capture spatial aspects, such as sprite shape, which change in dependency of the inferred cause. After training the network and freezing weights, new generalized observations $\tilde{o}$ can be encoded via iterative inference. The inferred stated can then be used for dynamical prediction of future timesteps, by applying $x' = f(z), x'' = f_z(z'), \ldots$. Figure 4 a) shows a typical extrapolation for up to 50 with different step sizes $dt$ scaling the predicted

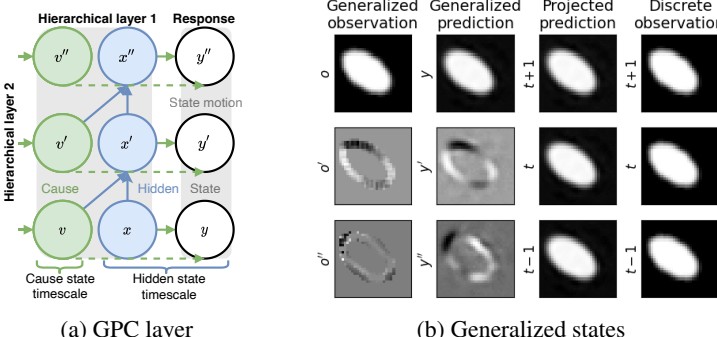

(a) GPC layer

(b) Generalized states

Figure 3: a) Hierarchical predictions (green) express expectations about causes (or data) in the next lower layer. Dynamical predictions (blue) predict higher orders of state motion. Dotted connections indicate optional skip connections between causes and the layer's hierarchical response (not used here). b) Discrete video frames (right) are fed to the model as generalized observations (left). The generalized sensory prediction of the network can be projected back to discrete sequences (center).

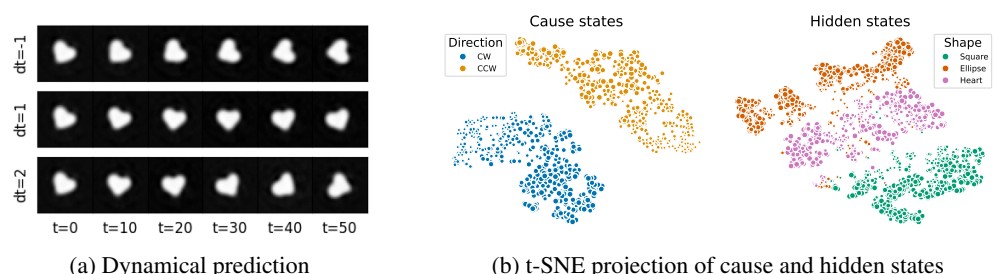

(a) Dynamical prediction

(b) t-SNE projection of cause and hidden states

Figure 4: a) Dynamical prediction with learned causes over three different step sizes $dt$. Shown is every tenth of 50 steps. b) t-SNE projection of cause and hidden states after unsupervised learning. Hidden states encode spatial aspects, such as shape while cause states encode hidden state motion and cluster into rotation directions. Marker sizes indicate the scale of observed sprites.

hidden state motion $x' = f(x, v) * dt$. Changing the applied step size allows to increase and decrease the speed of motion forward and backward in time. Figure 3 b) shows the discrete observations and their temporal embeddings. Additionally shown are the model's generalized prediction $\tilde{y}$ from hidden states as well as the resulting inverse embedding to a discrete sequence.

## 7    Conclusion

We presented a generalized predictive coding network (GPC) that uses Hebbian updates and the Laplace approximation with nonlinear neural networks to infer posterior distributions. We have shown that the model performs comparably to VAEs trained with exact error backpropagation. We extended the model to cover dynamical predictions of simple video sequences and demonstrated the possibility to learn dynamics using generalized coordinates of motion. We found that in many cases, GPC still learns meaningful dynamics on video datasets with lower sampling rate. For this work however, we focus on high resolution data. Important steps for future work could be to use convolutional neural networks or a comparison to related dynamical models, such as Neural ODEs or RNNs [22, 19].

## 8    Acknowledgments

This research was funded by the Federal Ministry of Education and Research of Germany (BMBF) as part of the "Cognitive neuroscience Inspired techniques for eXplainable AI" (CogXAI) project.

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

## A    Inference and learning with precision weighted prediction errors

For a hierarchical layer l, the variational free energy $\mathcal{F}$ decomposes into an accuracy term, that measures the quality of the outgoing prediction $g^{(l)}(\mu^{(l)})$, and a complexity term between top-down predicted state $g^{(l+1)}(\mu^{(l+1)})$ and inferred state $\mu^{(l)}$. Since we want to compare the GPC model to variational autoencoders, we additionally regularize by the distance between a standard normal prior distribution and the inferred posterior distribution:

$$\mathcal{F}(o, q^{(l)}, \hat{q}^{(l)}) = \mathbb{E}_q[\log p(o \mid z)] - \mathrm{KL}(q^{(l)}(z)\|\hat{q}^{(l)}(z)) - \mathrm{KL}(q^{(l)}(z)\|p(z)) \tag{1}$$

where $q^{(l)}(z)$ is the inferred distribution, $\hat{q}^{(l)}(z)$ is its top-down prediction and $p(z)$ is the chosen prior distribution. Hidden hierarchical layers predict the activity $\mu^{l-1}$ in the layer below instead of sensory observations $o$.

A central idea in predictive coding is to approximate the KL divergences in 1 with respect to prediction errors $\epsilon$:

$$\begin{aligned}
\epsilon^{(l,p)} &= (\mu^{(l)} - 0) \\
\epsilon^{(l,v)} &= (\mu^{(l)} - g^{(l+1)}(\mu^{(l+1)})) \\
\epsilon^{(l,o)} &= (g^{(l)}(\mu^{(l)}) - o)
\end{aligned} \tag{2}$$

where $\epsilon^{(l,o)}$ measures the reconstruction accuracy, $\epsilon^{(l,v)}$ measures the distance between inferred activity $\mu^{(l)}$ and top-down predicted activity $\hat{\mu}^{(l)}$ and $\epsilon^{(l,p)}$ measures the distance of inferred activity from the prior with zero mean. To make use of amortized inference, at the start of iterative inference, the inferred posterior is initialized with its top-down prediction $\mu_0 = \hat{\mu}$. During inference, the optimal $\mu^{*(l)}$ with respect to complexity and accuracy is computed using a gradient descent on $\epsilon^{(l,p)}$ and $\epsilon^{(l,o)}$. Under the Laplace approximation, the divergence from a standard normal prior simplifies to the divergence from zero mean.

After inference, the covariance parameters of the top-down predicted distribution $\hat{q}(z)$ and the inferred posterior distribution $q^*(z)$ are inferred following the routine described in Park et al. [16] using $\Sigma^{-1} = -\nabla_z^2 \log p_\theta(o, z)\big|_{z=\mu}$, which can be efficiently computed for ReLU activations. Then, the weights of the model can be updated with respect to the distance between optimal inferred distribution and the top-down predicted distribution:

$$\mathcal{F}(o, q^{(l)}, \hat{q}^{(l)}) = \mathbb{E}_q[\log p(o \mid z)] - \mathrm{KL}(q^{*(l)}(z)\|\hat{q}^{(l)}(z)) \tag{3}$$

where the KL divergence is approximated by a precision weighted prediction error:

$$\xi^{(l,v)} = \Sigma^{(l,v)^{-1}} \epsilon^{(l,v)} \tag{4}$$

We train the weights of all predictive coding models using the weighted prediction errors. For model evaluation and the comparison to VAE and VLAE baselines, we use the full (analytical) KL divergence between prior and posterior. We found that using precision weighted state updates of form $\dot{\mu} = \left(W_z^T W_z + I\right)^{-1} W_z^T \varepsilon \mid_{z=\mu}$ during iterative state inference improves performance in the dynamical model, when there is no top-down amortization (i.e. a single hierarchical layer is used). In the current configuration, the ELBO of static models does not improve with such weighted inference updates.

## B    Generalized coordinates from discrete sequential data

We compute temporal embeddings of observations according to a Taylor expansion of form

$$f(x \pm dx) = f(x) \pm \mathrm{dx} f'(x) + \frac{dx^2}{2!} f''(x) \pm \frac{dx^3}{3!} f'''(x) + \frac{dx^4}{4!} f''''(x) \pm \ldots \qquad (5)$$

for points $x \pm dx$ around a point $x$ assuming a fixed step size e.g. $dx = 1$. Since we observe discrete samples $[o_1, ...o_n]$ we approximate the instantaneous derivatives $f', f'', ...$ up to desired order using a central finite difference operator $\delta_d x^n[f](x)$. We interpret the resulting differencing coefficients as convolutional kernels, which can be applied to any sequential data with sufficiently high sampling rate either online or during preprocessing. Mapping back from the network's states to sequential data can easily be done using the inverse kernel. Figure 5 shows examples for forward and inverse embedding kernels.

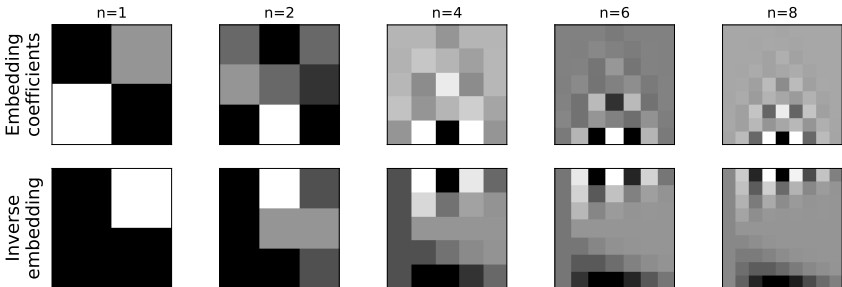

Figure 5: Forward and inverse embedding kernels for five different embedding orders.

## C    Static predictive coding

### C.1    Influence of iterative inference steps

|       | MNIST    | OMNIGLOT | fMNIST   | MNIST    | OMNIGLOT | fMNIST   |
|-------|----------|----------|----------|----------|----------|----------|
| Steps | 3        | 3        | 3        | 6        | 6        | 6        |
| PC_S  | 49.0±0.3 | 44.9±1.0 | 42.8±0.9 | 48.4±0.2 | 60.2±3.9 | 36.3±0.3 |
| PC    | 48.9±0.3 | 49.0±0.3 | 37.1±0.2 | 48.2±0.1 | 56.1±1.4 | 36.8±0.1 |
| VAE   | 46.6±0.5 | 41.6±0.6 | 36.5±0.2 | 46.6±0.5 | 41.6±0.6 | 36.5±0.2 |
| VLAE  | 46.7±0.1 | 42.1±0.1 | 38.1±0.2 | 47.2±0.1 | 42.5±0.1 | 38.8±0.1 |

Table 4: Complexity (test set)

Table 4 shows the posterior complexity of models trained on the static prediction task for MNIST, OMNIGLOT and Fashion MNIST in terms of mean and standard deviation over five runs. The predictive coding models GPC-S and GPC-M show complexity that is comparable to, but slightly higher than the complexity of the baseline VAE. For VLAE, increasing the amount of inference steps

from 3 to 6 slightly increases the complexity of encoded states. For GPC models, increasing the amount of inference steps sometimes leads to an improvement in terms of complexity.

## C.2 Influence of iterative inference steps

We found that, for static models, increasing the amount of inference steps is beneficial primarily for low amounts (1-5 steps). Figure 6 show test set results on the MNIST dataset with various amounts of iterative inference steps. Since the top-down prediction performs amortized inference of the inferred state, even a single inference step leads to meaningful results. Choosing a higher inference update rate (e.g. 0.01) often shows favorable performance over choosing a lower learning rate (e.g. 0.001) with more update steps.

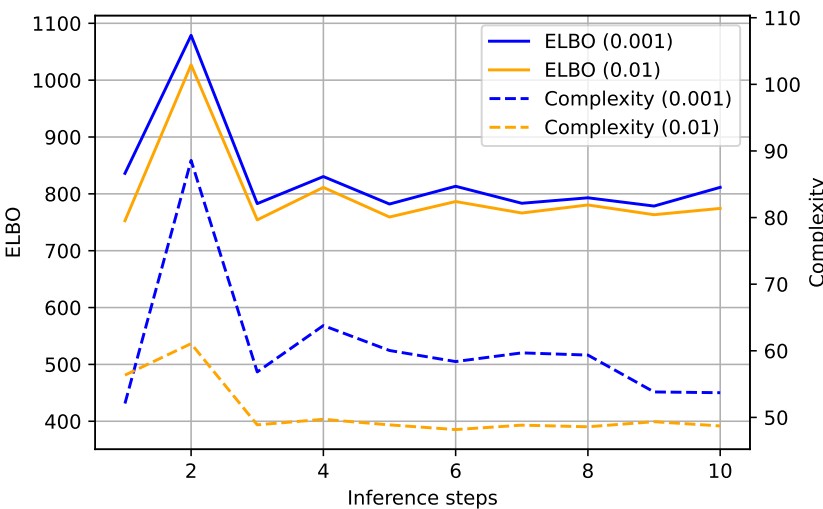

Figure 6: Evidence lower bound and complexity on the MNIST test set after training with different amounts of iterative inference steps. Shown are two different runs with iterative inference update rates (0.01 and 0.001).

## D  Data sources

We employ three popular datasets for unsupervised learning on images: The MNIST dataset (Creative Commons Attribution-Share Alike 3.0 license), Fashion MNIST (MIT license) and OMNIGLOT (MIT License). Evaluation of the dynamical predictive coding model is based on the Disentanglement testing Sprites dataset (Apache License 2.0). Table 5 shows sources and licenses for all datasets. MNIST and FashionMNIST contain 60000 train and 10000 test images (with $28 \times 28$ pixels) while the OMNIGLOT dataset contains 24345 train and 8070 test images (also with $28 \times 28$ pixels).

| Dataset | Source | License |
|---------|--------|---------|
| MNIST [12] | yann.lecun.com/exdb/mnist | CC BY-SA 3.0 |
| OMNIGLOT [11] | github.com/brendenlake/omniglot | MIT license |
| Fashion MNIST [24] | github.com/zalandoresearch/fashion-mnist | MIT license |
| dSprites [14] | deepmind.com/open-source/dsprites | Apache License 2.0 |

Table 5: Data sources

To generate discrete video sequences with high temporal resolution for the dynamical GPC model we use a customized version of the dSprites dataset [14]. We generate 128000 random samples from the original dataset and apply Gaussian blur along both spatial axes with kernel size 3 and Standard

deviation of 10 before applying normalization. We restrict x and y positions to six values respectively, such that all sprites appear within a center crop of 32 x 32 pixels. Starting with the noisy version of the sprite, we apply a single direction of rotation (counterclockwise) by rotating the sprite by a single degree. All remaining aspects, such as shape, size, horizontal position and vertical position stay constant. The Gaussian noise was applied only to the first frame of each sequence. We then projected the resulting discrete video sequences into generalized coordinates using the embedding kernels discussed in Section B of the appendix.

