# OpenReview forum: "Generalized Predictive Coding: Bayesian Inference in Static and Dynamic models"
_NeurIPS.cc/2022/Workshop/SVRHM — SVRHM Poster_

### Official Review · Reviewer_4MmX · 2022-10-16

**Rating:** 5
**Confidence:** 2

**Review:**

This paper proposed an empirical comparison of predictive coding models (which may be more biologically plausible) with variational autoencoders. They also evaluated dynamic predictive coding models on a variant of the DSprites dataset. Overall, I felt like the question, approach, and evaluation were valuable, though I have some concerns.

1. Does the approach of performing local inference scale to deeper architechtures, where updates need to be performed at the multiple levels? In the current implementation, it appeared that the local update was only applied to the first layer, as the second was fixed.

2. The empirical evaluation on Mnist/FashionMnist/Dsprites are somewhat toy for a benchmarking study -- perhaps it would be valuable to consider more challenging image/video dasasets (such as CelebA), to better understand the comparisons on more realistic/challenging tasks?

3. As someone who does not work directly with predictive coding, I appreciated the effort the authors spent to keep the paper self contained, though I believe the authors can more clearly disentangle their contribution from prior work (i.e. it was not clear to be if nonlinear predictive coding models had previously been implement, or if they simply had not been applied to image classification tasks). I also felt that sometimes notation was used that was not defined, which made the paper more difficult to follow. For example:

  - VFE: acronym not defined
  - The different GPC models were not specified, as to how they differed (GPC-V, GPC-M).
  - (L49-50): When talking about the distribution, it could have been more clear that they  were referring to the conditioned variable.
  - Should line 67 be v+1 for the next higher layer (instead of v-1)
  - \Sigma^{(l)v-1} not directly defined.

---

### Official Review · Reviewer_ct4J · 2022-10-16
**Imporant first steps in comparing predictive coding and deep generative models**

**Rating:** 8
**Confidence:** 3

**Review:**

This is an important paper, linking an influential theory from cognitive neuroscience to the machine learning field of deep generative models (variational autoencoders in particular). Deep generative models have shown impressive advances in machine learning. Those models are data-hungry and their training and implementation have an elusive relation to biological learning and circuits. Predictive coding has been highly influential in cognitive neuroscience and touches upon several questions that are highly relevant to our understanding of the brain: How can hierarchical generative models be learned using only local learning rules? How is model inversion implemented in the brain's dynamics? And how are beliefs about dynamics in the world best represented in neural states? However, it is still unclear to what extent the predictive coding principles actually scale to real-world environments and inference. The current paper, therefore, is an important starting point for bringing both kinds of models together.

The paper is clear and very well written. I do have some remarks that might be helpful feedback for future presentations of this work.

1) The audience might not be deeply familiar with Generalized Predictive Coding (Friston, 2008). Didactically, it would be helpful to introduce the rationale and relevant concepts a bit more thoroughly to the reader (e.g., hidden vs. cause states; generalized coordinates of state motion). Admittedly, the space for the submission was limited, but the authors might want to consider this for future submissions, in particular, if parts of the audience are mainly exposed to the deep generative model literature.

2) There has been quite a bit of recent work in combining deep generative models with predictive coding (e.g., Lotter et al., 2016; Han et al., 2018; Spratling, 2019, Ofner & Stober, 2021). These approaches might not exactly match the flavor of the Fristonian conceptualization of predictive coding but it is still important to discuss this work’s relation to those other approaches.

3) The section on dynamical predictive coding is not quite as conclusive and integrated with the rest of the paper. What is the rationale for comparing the GPC-L1 and GPC-all models? Previous work has found that minimizing the prediction error only in the lowest layer rather than in all layers leads to the best-performing model (Lotter et al., 2016). However, this finding seems to be at odds with the overall principle of predictive coding. I was also hoping to find quantitative comparisons to other models in the dynamical predictive coding section.

4) Why is the VLAE consistently better than the GPC models? It would be great to understand which differences between the VLAE and the GPC models are most likely to produce the performance gap. In general, I would have appreciated a more thorough discussion of the results. E.g., does the GPC model outperform the VAE because of a lower amortization or approximation error (Cremer et al., 2018)? Does the VLAE outperform the GPC because of a more favorable learning regime?

References:

- Lotter, Kreiman, Cox (2016). Deep Predictive Coding Networks for Video Prediction and Unsupervised Learning. http://arxiv.org/abs/1605.08104
- Han, Wen, Zhang, Fu, Clurciello, Liu (2018). Deep Predictive Coding Network with Local Recurrent Processing for Object Recognition. http://arxiv.org/abs/1805.07526
- Spratling (2019). A Hierarchical Predictive Coding Model of Object Recognition in Natural Images. Cogn Comput 9 (2)
- Ofner & Stober (2021). Differentiable Generalised Predictive Coding. https://arxiv.org/abs/2112.03378
- Cremer, Li, & Duvenaud (2018). Inference Suboptimality in Variational Autoencoders. http://arxiv.org/abs/1801.03558

---

### Official Review · Reviewer_DMSV · 2022-10-16
**Interesting work but is of limited practical use**

**Rating:** 5
**Confidence:** 3

**Review:**

The paper proposed a generalized predictive coding model, compared the static model with VAE and VLAE on image tasks, and showed the experiment result with log-likelihood value. The dynamic model is tested on sprites datasets, but no comparison with existing models. This is not a well-written paper. The author did not clearly state what is new in their model in the main part until the description in sections 3.2 and 3.3. The author did not give enough literature review on predictive coding and VAEs and did not describe clearly the main contribution and innovation of their models. What’s more, there’s limited creativity in their model.

The author proposed a generalized PCN, combining the technique of the VLAE and multilayer nonlinear network. However, the performance is still not good, which shows no practical value for the work. There are lots of notions that are not clearly identified and explained. The author should compare their model with recent works like hybrid productive coding [1] and PredNet [2]. The predictive coding combined with CNN and RNN has been done in previous works.

Line 25, there should be (VFE) after the free energy function since VFE appears on line 65 and line 74.

Line 90, should place [16] where VLAE is mentioned in the paper for the first time.

On page 3, Figure 1, there’s no description for GPC-V and GPC-M, what the V and M stand for, which appears on line 133.

On page 5, Table 2, what are GPC-all and GPC-L1 stand for? There’s no comparison for GPC-all, GPC-L1 with any existing model.

On page 8, between line 271 and line 272, formula (1), should be the kth power of x-a, and k should be in the superscript position.

On page 8, Table 3, and page 10, Table 5, there’s no reason to omit the value for VAE with 6 steps.

On page10, Table 5, the ELBO for GPC is worse than VAE and VLAE, which shows this model is not practical.

---

### Official Review · Reviewer_JJLr · 2022-10-17
**Good paper**

**Rating:** 7
**Confidence:** 3

**Review:**

Summary :

The authors first observe that the Predictive Coding Networks (PCNs) and variational autoencoders (VAEs) have striking similarities but lack a quantitative comparison. They later propose a generalised PCN that parametrises the latent distributions under Laplace approximation. With some proof of concept experiments on small datasets, they find that their GPC networks are comparable in performance.


Main Review :

The strength of the article lies in its contribution of a novel proposal. At the same time, the weakness lies in its inability to convincingly demonstrate its utility. But, given the early stage of this direction, I believe it is okay. Overall, I believe the proposal is quite intriguing and will be a nice addition to the SVRHM discussions.

- The authors should have a look at [1] which already derived the correspondence between VAEs and PCNs. I think that work is directly relevant.

- Since authors intend to extend their work to convolutional networks or LSTMs, they can look at the publicly available code of [2] (for LSTMs) and [3] (for CCNs).




[1]- Boutin, Victor, et al. "Iterative VAE as a predictive brain model for out-of-distribution generalization." _arXiv preprint arXiv:2012.00557_ (2020).

[2] - Lotter, William, Gabriel Kreiman, and David Cox. "Deep predictive coding networks for video prediction and unsupervised learning." _arXiv preprint arXiv:1605.08104_ (2016).

[3] - Choksi, Bhavin, et al. "Predify: Augmenting deep neural networks with brain-inspired predictive coding dynamics." _Advances in Neural Information Processing Systems_ 34 (2021): 14069-14083.